# Companion Animal Ownership and Mood States of University Students Majoring in Animal Sciences during the COVID-19 Pandemic in Japan

**DOI:** 10.3390/ani11102887

**Published:** 2021-10-03

**Authors:** Daiki Namekata, Mariko Yamamoto

**Affiliations:** Department of Animal Sciences, Teikyo University of Science, Yamanashi 409-0193, Japan; daikinamekata2021@gmail.com

**Keywords:** companion animal ownership, university students, COVID-19, attachment, POMS2

## Abstract

**Simple Summary:**

People have experienced great difficulties in their daily lives from the COVID-19 pandemic. This study examines whether living with companion animals and attachment to companion animals influence the moods of university students in Japan. In this study, students answered a questionnaire regarding their demographic data, companion animal ownership, attachment to their companion animals, perceived difficulties from COVID-19, and mood states. The results indicated that companion animal ownership with high attachment to their companion animals would relate to a positive mood in university students majoring in animal sciences during the COVID-19 pandemic. However, because of the limited population in size and by the students’ major, the results need to be interpreted as a possible effect of companion animals, and not as conclusive evidence to support the effects of animals.

**Abstract:**

COVID-19 caused great difficulties in many people’s daily lives, including university students in Japan. This study examined whether living with companion animals and attachment to companion animals influence the moods of university students. Students answered a questionnaire, including demographic data, companion animal ownership, attachment to their companion animals, perceived difficulties from COVID-19, and Profile of Mood States 2 (POMS2) results. A total of 180 students answered the questionnaire. Stepwise multiple regression analyses were conducted to identify factors related to the total mood disturbance (TMD) score of the POMS2. In the regression model, perceived difficulties from COVID-19 and having a companion animal and a strong attachment to their companion animals were significantly correlated with TMD and served as the predictor variables. The first variable was positively related to TMD, whereas companion animal ownership with high attachment to their companion animals was negatively related to TMD. This finding indicated that companion animal ownership with high attachment to their companion animals would relate to a positive mood in university students majoring in animal sciences during the COVID-19 pandemic. However, because of the limited population in size and by the students’ major, the results need to be interpreted as a possible effect of companion animals, and not as conclusive evidence to support the effects of animals.

## 1. Introduction

During the COVID-19 pandemic, the number of new companion animal owners increased in Japan, which was similarly reported in other countries, such as the US and UK [1,2]. According to the annual survey of the Japan Pet Food Association conducted in October 2020, the number of companion dogs and cats within the previous year increased by 114% and 116% compared with in 2019, respectively [3]. In a study conducted in December 2020 on 1000 companion animal (cat and/or dog) owners, the participants were asked about their reasons for starting to live with a dog and/or a cat [4]. The percentage of people who chose “to spend enjoyable time at home” was 61.9% among people who started to live with their companion animals before the pandemic (March 2020) and 74.7% among people who started to live with their companion animals after the pandemic (April 2020). Increased time spent at home after the promotion of remote work and stay-at-home requests by the government may have facilitated the decisions of people to live with a companion animal. Approximately half of people chose “I think a companion animal will snuggle up when I’m lonely”, but the percentages did not differ among people who started to live with a companion animal before and after the pandemic (before: 46.4%; after: 43.7%) [4].

Companion animals can provide a great benefit to people living together. Although studies have not shown only the positive effects of living with animals, as explained later, many have indicated that people who live with a companion animal are likely to have good mental health [5,6]. Physical and social benefits of companion animal ownership have also been reported. Companion dogs can facilitate people’s physical activity, and people who live with a dog are likely to engage in a high level of physical activity [7,8]. Dog walking also leads to social interaction. Wood et al. reported that companion animal ownership provides potential opportunities for interactions between neighbors [9].

Companion animals also decrease loneliness. Kaneko and Murakami [10] reported that university students living or who have lived with a companion animal had significantly lower feelings of loneliness compared with university students without a companion animal. Antonacopoulos and Pychyl [11] showed that dog walkers who conversed with people they encountered while dog walking were less lonely than those who did not converse with people they encountered. However, reports about the relationship of living with a companion animal and loneliness are not always consistent. Systematic reviews of companion animals and loneliness determined that the inconsistency of results from previous studies and their research is caused by poor quality, especially for older ones [12,13]. Therefore, whether living with a companion animal can decrease people’s loneliness remains inconclusive. The inconsistency of benefits from animals is also reported among other known benefits, such as those related to physical and mental health and pet ownership, including cardiovascular disease, depression and anxiety [14,15]. In addition to the quality of research, the complexity of human–animal interactions makes it difficult to understand the study results. The benefits from companion animals are not simply gained from the existence of companion animals, but rather the bond and closeness with a companion animal, such as our interaction with them and relation to each other, may influence the effects of companion animals [16,17].

The COVID-19 pandemic has wreaked havoc worldwide and greatly changed people’s lives, including those of university students. A government study conducted in mid-May 2020, during the first declaration of a state of emergency (7 April–25 May 2020), showed that 90.0% of all Japanese universities and colleges were providing only online teaching, 3.1% were providing only in-person teaching, and 6.8% were providing a hybrid of in-person and online classes [18]. No detailed research has been completed on this to date, but it was reported that many universities initiated a campus closure during the first state of emergency, which was right after the school year began (the first semester typically starts on 1 April) [19]. Even in the second semester, 90.3% of the universities in the Kanto district (consisting of Tokyo and five other prefectures), where 33% of the population live, held hybrid in-person and online courses, and only 8.8% were providing in-person teaching for all classes [20]. Among the universities using a hybrid approach, 32.7% were providing classes mainly online. Although government data on the types of teaching offered by universities for the year 2021 have not yet been reported, more universities seem to have increased their ratios of in-person teaching [21]. During the COVID-19 pandemic and the subsequent great changes in teaching style, some studies have reported that university students were experiencing new burdens and psychological problems [22,23].

This study aims to investigate whether living with a companion animal and the level of attachment to their companion animals influence the mood states of university students majoring in animal sciences under the COVID-19 pandemic, especially around the time of the first outbreak of the disease, which occurred as students were also adjusting to life in a new academic year.

## 2. Materials and Methods

### 2.1. Participants

University students majoring in animal sciences at Teikyo University of Science were surveyed. The study was announced on an online bulletin board of some classes; 314 students, mainly in the first three years of their studies, had access to the bulletin board. The students come from all parts of Japan, and about half of them were living with their families. The university campus is located roughly 1–1.5 h by train from central Tokyo. These students were recruited for two reasons. First, this population made it possible to recruit enough pet owners. The overall rate of Japanese households’ pet ownership is only 28.2% (dogs: 11.9%, cats: 9.6%) [3], and we hypothesized that university students, especially those living alone, were less likely to live with a companion animal. Therefore, we focused on the students majoring in animal sciences, as they seemed more likely to have a companion animal. Second, we recruited only students from similar school situations. The situations of other universities were not well understood at the time when this study was being prepared, because it was just after the new school year had started and when the first declaration of a state of emergency was issued. Japanese universities were in the midst of a chaotic situation. Therefore, we only focused on students from one university, with which the authors are affiliated. The survey was conducted between 8 June and 5 July 2020, immediately after the first declaration of state of emergency was lifted. However, the campus was still closed, and classes were provided remotely.

This survey was conducted with anonymity and voluntary participation. The study adhered to the Declaration of Helsinki and was approved by the Ethics Committee of Teikyo University of Science.

### 2.2. Questionnaire

The questionnaire is available in the Appendix A. Questions on simple demographics pertained to age, gender, companion animal ownership and types of animals they own, living with someone or alone, and whether they were returning home (staying with family members) when answering the questionnaire. Participants with companion animals were also asked to answer the Companion Animal Attachment Scale (CAAS) [24]. The CAAS is used to measure Japanese attachment to a companion animal. The CAAS was created by Hamano [24] because no attachment scale is unique to the different Japanese history, culture, religion, thoughts toward animals, and lifestyle from the U.S. and European countries. The scale has 34 items on a five-point Likert scale.

Questions of perceived difficulties from the COVID-19 pandemic, ways of relieving stress, and mood states were also included. As for the perceived difficulties brought on by the COVID-19 pandemic, the students were asked one question: whether or not they were experiencing difficulties due to the effects of COVID-19, and they were asked to rate with a four-point Likert scale from 1, 2, 3, and 4 as no difficulty, slight difficulty, moderate difficulty, and great difficulty, respectively. The ways of relieving stress, including watching TV, watching movies, reading books and comics, watching online videos, checking a social networking service (SNS), playing games, playing with a companion animal, sleeping, talking with family members (making a phone call to family members), making a phone call to friends, exercising, and others were in multiple choices. These activities were selected based on research in which students noted their preferred ways of relieving stress [25,26] and some additional activities, such as checking SNS and playing with a companion animal. These were not standardized questions aimed at measuring the students’ stress management skills, but rather questions aimed at determining what kinds of activities the students use to relieve their stress. The mood states were scored using the short version of the Profile of Mood States second edition (POMS2) short form [27]. POMS2 short form is a mood inventory used to assess transient feelings and mood. The form contains 35 adjectives of a five-point Likert scale (0—not at all to 4—extremely) that describe seven different moods, namely, “anger–hostility (AH)”, “confusion–bewilderment (CB)”, “depression–dejection (DD)”, “fatigue–inertia (FI)”, “tension–anxiety (TA)”, “vigor–activity (VA)”, and “friendliness (F)”. Participants answered their mood states during the previous one-week period. The total mood disturbance (TMD) score and the seven subscales indicated the participants’ mood states. The T-scores for the TMD score and seven subscales were used for the analyses in this study.

The participants were informed that the survey aimed to investigate their mood states during the COVID-19 pandemic and that the main purpose of investigating whether their moods would be influenced by companion animal ownership and their attachment to their companion animals was not disclosed.

### 2.3. Companion Animal Ownership and Attachment to Their Companion Animals

The participants were divided into three groups: 1—not companion animal owners, 2—companion animal owners with low attachment to their companion animals, and 3—companion animal owners with high attachment to their companion animals. The low and high attachment groups were divided by the average CAAS score among the participants.

### 2.4. Statistics

Using IBM SPSS Statistics 23, a multiple comparison test with Bonferroni correction was used to determine the differences between companion ownership/attachment to their companion animals and the students’ perceived difficulties from the COVID-19 pandemic. The Mann–Whitney U test was used to check the differences in levels of attachment to a companion animal, based on gender and type of animal. Stepwise multiple regression analyses were conducted to identify factors related to the TMD scores and seven subscales.

## 3. Results

### 3.1. Participants’ Characteristics

Table 1 summarizes the respondents’ demographic characteristics by gender and companion animal ownership/attachment. The total sample size was 180. The response rate in this survey was 57.3%. Survey respondents were 124 women (68.9%) and 56 men (31.1%). The average age was 19.4 years old.

At the time of answering the questionnaire, 128 students were living with their family members or friends. Among them, 54 students were originally living alone but returned home due to the COVID-19 pandemic and school closure. A total of 52 students were living alone at the time of answering the questionnaire.

### 3.2. Perceived Difficulties from the COVID-19 Pandemic and Ways of Relieving Stress

Most students (82.2%) felt some difficulties from the COVID-19 pandemic. As ways of relieving stress, reading books and comics, watching online videos, checking SNS, playing games, playing with a companion animal, and sleeping were chosen by more than half of the students. The students selected an average of 5.4 ways of relieving stress.

### 3.3. Companion Animal Ownership and Attachment to Their Companion Animals

Approximately half of the students had a companion animal(s) (*n* = 92, 51.1%). Among them, 66.3% had a dog(s), 25.0% had a cat(s), 18.5% had a small animal(s), and 26.1% had other types of animals, such as fishes and birds.

The average attachment score was 128.8 (S.D. = 22.7). Among companion animal owners, 43 and 49 students were grouped in the low (attachment score < 128.8) and high attachment groups (attachment score > 128.8), respectively.

The Mann–Whitney U test showed that the attachment score was significantly higher among women than men (women: mean = 132.5, S.D. = 22.1; men: mean = 119.9, S.D. = 22.1, *p* < 0.05, *r* = 0.25). When the type of animal was indicated in detail, a gender difference was only revealed among dog owners (women: mean = 141.4, S.D. = 17.9; men: mean = 126.2, S.D. = 16.6, *p* < 0.01, *r* = 0.38). Similarly, dog owners scored significantly higher for attachment, compared to other pet owners among women (dog owners: mean = 141.4, S.D. = 17.9; other pet owners: mean = 116.3, S.D. = 20.0, *p* < 0.01, *r* = 0.54). No statistical difference was shown among men (dog owners: mean = 126.2, S.D. = 16.6; other pet owners: mean = 105.0, S.D. = 27.3, *p* = 0.06, *r* = 0.36).

### 3.4. Influence of Companion Animal Ownership and Attachment to Their Companion Animals on Perceived Difficulties from the COVID-19 Pandemic

A multiple comparison test with Bonferroni correction was used to determine whether the students’ perceived difficulties from the COVID-19 pandemic differed depending on the companion animal ownership and attachment to their companion animals.

No statistical differences were found in their perceived difficulties from the COVID-19 pandemic among the following groups: no companion animal owners, low attachment group, and high attachment group (*p* > 0.05).

### 3.5. Factors Affecting Mental States

Stepwise multiple regression analyses were conducted to identify factors related to the TMD score and the subscales of the POMS2. In the regression model, each POMS2 score was used as the outcome variable. Other items, such as age, gender, living with someone or alone, companion animal ownership/attachment (no ownership, low attachment, and high attachment), types of companion animals, perceived difficulties, and ways of relieving stress, were used as predictor variables.

#### 3.5.1. TMD

When the TMD score was used as the outcome variable, perceived difficulties from the COVID-19 pandemic and companion animal ownership/attachment were significantly correlated with TMD and served as the predictor variables (*R*^2^ = 0.098; *F* = 10.722, df = 2, *p* < 0.001). Perceived difficulties from the COVID-19 pandemic were positively related to TMD (β = 0.281, *p* < 0.001), whereas companion animal ownership/attachment was negatively related to TMD (β = −0.165, *p* < 0.05) (Table 2). Students with more perceived difficulties from the COVID-19 pandemic were more likely to have higher TMD scores (worse mood), whereas companion animal owners with higher attachment to their companion animals were more likely to have lower TMD scores (better mood).

#### 3.5.2. Subscales of POMS2

Each subscale of POMS2 was used as the outcome variable in the regression analyses. The details are shown in Table 2.

When the AH score was used as the outcome variable, perceived difficulties from the COVID-19 pandemic was significantly correlated with AH and served as the predictor variables (*R*^2^ = 0.069; *F* = 14.175, df = 1, *p* < 0.001).

When the CB score was used as the outcome variable, perceived difficulties from the COVID-19 pandemic, companion animal ownership/attachment, and number of ways to relieve stress were significantly correlated with CB and served as the predictor variables (*R*^2^ = 0.088; *F* = 6.731, df = 3, *p* < 0.001).

When the DD score was used as the outcome variable, perceived difficulties from the COVID-19 pandemic were significantly correlated with DD and served as the predictor variables (*R*^2^ = 0.064; *F* = 13.289, df = 1, *p* < 0.001).

When the FI score was used as the outcome variable, perceived difficulties from the COVID-19 pandemic, companion animal ownership/attachment, and number of ways to relieve stress were significantly correlated with FI and served as the predictor variables (*R*^2^ = 0.071; *F* = 5.562, df = 3, *p* < 0.001).

When the TA score was used as the outcome variable, perceived difficulties from the COVID-19 pandemic, gender, number of ways to relieve stress, and dog ownership were significantly correlated with TA and served as the predictor variables (*R*^2^ = 0.120; *F* = 7.086, df = 4, *p* < 0.001).

When the VA score was used as the outcome variable, perceived difficulties from the COVID-19 pandemic and companion animal ownership/attachment were significantly correlated with VA and served as the predictor variables (*R*^2^ = 0.055; *F* = 6.170, df = 2, *p* < 0.001).

When the F score was used as the outcome variable, gender was significantly correlated with F and served as the predictor variables (*R*^2^ = 0.033; *F* = 7.131, df = 1, *p* < 0.001).

Overall, perceived difficulties from the COVID-19 pandemic were positively related to AH, CB, DD, FI, and TA and negatively related to VA. These results show that students with more perceived difficulties from the COVID-19 pandemic were more likely to have worse scores in the subscales of negative and positive moods. The number of ways to relieve stress was positively related to CB, FI, and TA. This finding shows that those trying various things to relieve their stress were more likely to have worse scores in the subscales of negative moods. Gender (women) was negatively related to TA and F, and women were more likely to have better scores in TA and worse scores in F.

Dog ownership was negatively related to TA, and dog owners were more likely to have better scores in TA. Furthermore, companion animal ownership/attachment was negatively related to CB and FI and positively related to VA. These results show that companion animal owners with higher attachment to their companion animals were more likely to have better scores in CB, FI, and VA.

## 4. Discussion

This study aimed to investigate whether living with a companion animal and the level of attachment to companion animals influence the mood states of university students majoring in animal sciences during the COVID-19 pandemic in Japan. The multiple regression model showed that companion animal ownership/attachment served as a predictor variable for the outcome variables of TMD (total mood disturbance), CB (confusion–bewilderment), FI (fatigue–inertia), and VA (vigor–activity). Perceived difficulties from the COVID-19 pandemic served as a predictor variable for all outcome variables, except F. The number of ways of relieving stress, gender, and dog ownership also served as predictor variables for some outcome variables. The adjusted R^2^ s were small, which were between 0.033 and 0.120. However, under the limited interaction with other people under the stay-at-home request, companion animal ownership/attachment and dog ownership, but not whether living with someone/alone or returning home (staying with family members), were selected as the predictor variables that influenced the students’ mental states.

No differences were found between non-companion animal owners and companion animal owners with low and high attachment to their companion animals on the level of perceived difficulties from the COVID-19 pandemic. This finding indicates that companion animal ownership and having strong attachment to their companion animals seem to not mitigate the owners’ perceived difficulties from the hard situation. Instead, this finding indicates that even if people feel the same level of difficulties, the existence of companion animals and interaction with companion animals caused by strong attachment to their companion animals may stabilize the mental states of the students. Indeed, among the companion animal owners (*n* = 92), 89.1% (*n* = 82) answered that they played with a companion animal to relieve their stress. This result shows that most companion animal owners recognize that their companion animals help them relieve their stress.

People with higher attachment to their companion animals go for a walk with their dogs and take care of their companion animals more often than people with lower attachment to their companion animals [28,29]. In this study, 66.3% (*n* = 61) were dog owners, and the exercise requirements of their dogs may increase the owners’ physical activity, as reported in studies on dog walking [7,8]. Other types of companion animals, which do not require walks, also require daily care and can help owners maintain an orderly life. Therefore, people with higher attachment to their companion animals and providing adequate exercise and care for their companion animals may have received a positive influence regarding managing their health and keeping an orderly life even during the stay-at-home request. The influence of companion animal ownership under the COVID-19 pandemic has been extensively reported. In a large study in the UK, Ratschen et al. showed that companion animal ownership, in comparison with non-ownership, was associated with smaller decreases in mental health and smaller increases in loneliness since the lockdown [30]. Our results support this research result, and companion animal ownership and interaction with companion animals would have contributed to maintaining people’s mental status during the pandemic. Another study conducted in Australia showed that dog ownership and high levels of mindfulness protect against loneliness during a lockdown, but cat ownership shows a different result [31]. Similarly, in our study, among several types of companion animals, only dog ownership served as a predictor variable in TA (tension–anxiety). The exercise requirement for dogs and their nature of actively engaging with people may have positively affected owners’ mental status in difficult situations.

The results showed some gender differences. Women scored significantly higher in terms of attachment to their companion animals, especially dogs, when compared to men. This result is consistent with Winefield and her colleagues’ research, which also showed higher pet attachment among women than men, and the score was significantly higher for dog owners than for cat or other pet owners [32]. Although previous research on gender and pet attachment has been inconsistent, those studies reporting a relationship between gender and pet attachment always show that women have higher levels of attachment to pets than men do [33,34,35]. Interpreting the results of a survey conducted in Japan, Sugita reported that women spent more time with their pets than men. Furthermore, a positive correlation between attachment to dogs and the time spent with dogs was shown for both men and women [36,37]. In the present study, the time spent with pets was not asked; however, it is possible that female students spent more time with their pets, and as a result, they might have been more attached to their pets in comparison to male students. As discussed later, interactions with companion animals, especially dogs, might have served an important role for women under the limited social interactions allowed by the pandemic-related lockdown.

Gender also served as a predictor variable on F (friendliness) and TA. Gender was the only predictor variable, and women were more likely to have worse scores in F. The questions for F consisted of interpersonal items. Because people were required to engage in social distancing because of the COVID-19 pandemic, in-person social interaction decreased greatly. Previous studies have reported that women have larger social networks and receive support from more sources than men [38], and they were more likely to have friends or family to whom they could talk about their health. Women were also more likely to be a member of one or more community organizations [39]. Women might have been more affected by the great changes in social interaction than men were because of the COVID-19 pandemic, as shown in a study conducted in the UK wherein women reported greater degrees of loneliness than men during the pandemic [40].

As for TA, the women’s better scores in TA compared to men were the opposite of other findings on mental health during the COVID-19 pandemic. It was also a surprising result considering the worse F scores reported in women. Studies conducted in several countries, including Japan, have shown that being female is one of the risk factors for poor mental health [41,42]. One study focusing on medical students in Japan did not show gender differences in mental health [43]; however, more studies have indicated being female is a mental health risk factor. One possible reason for the current result might be explained by women’s higher levels of attachment to their dogs, as “dog ownership” was another predictor variable for TA. Having a positive relationship with a dog might have worked to maintain women’s mental health, even during the COVID-19 pandemic. Many studies have investigated the relationship between pet ownership and mental health, but there is only limited research specifically investigating the relationship between dog ownership and anxiety. One study showed that dog ownership relates to less anxiety [44], while another showed no relationship between dog ownership and anxiety [45]. These works did not investigate in detail the relationships people have with dogs. For example, one study on people diagnosed with fibromyalgia by Silva and colleagues reported that dog ownership was associated with reduced levels of anxiety in individuals who also have moderate to high levels of human social support, but the opposite was true for individuals with low levels of social support [46]. This study also showed that some interactions with dogs, such as petting and stroking the dog, which help to manage pain, are associated with a significant reduction in anxiety levels. This was so regardless of the levels of human social support. Similarly, a study on older adults showed that dog ownership was not associated with anxiety levels, but the frequency of a dog’s presence was moderately negatively related to anxiety [47]. Based on these results, dog owners who have a positive relationship with their dogs may have less anxiety.

Finally, the number of ways to relieve stress served as predictor variable in FI (fatigue–inertia), CB (confusion–bewilderment), and TA, and these scores were worse when students used more activities to relieve stress. Even during the pandemic, students had to participate in classes and hand in more reports than before the pandemic, which caused a great deal of stress [48]. Self-management was required of the students to adjust to the changes. In this situation, students who chose to engage in many activities to relieve stress may not have had an orderly life or may not have been able to focus on their schoolwork, which would have led to negative mental moods.

### Limitations

There are some limitations in this study. First, the study was not designed to investigate the causality between companion animal ownership/attachment and mental states of students. The mental states of companion animal owners might be originally better than those of non-companion animal owners. Second, whether this result is a unique result under the COVID-19 pandemic remains unclear. Even in the absence of the social turmoil of the COVID-19 pandemic, companion animal owners (those with a strong attachment to their companion animals) may be in good mental states. However, another study of university students reported that level of attachment to their companion animals was not significantly related to psychological quality of life [49]. Therefore, the effects obtained from a better relationship with companion animals might have become larger in a difficult situation, such as the COVID-19 pandemic.

This study was conducted voluntarily, and a small sample size was also a limitation. A larger sample size is required for a valid conclusion. However, given that the main purpose of the study was to investigate the effects of companion animals, the recruitment of this survey was not disclosed. Therefore, the bias of the responses of companion animal owners was minimized.

Applebaum et al. reported that some problems were experienced by companion animal owners in the COVID-19 pandemic [50]. Although the current study did not show the negative aspects of companion animals under the pandemic, the influence of companion animals on students could be investigated objectively by including questions asking about negative experiences.

Finally, this survey focused only on students majoring in animal sciences, who may have positive relationship with animals and appreciate the effect of animals more often than the students with other interests. It is possible that the study results were unique to students who have an interest in animals and that the results cannot be generalized to other university populations.

In the future, by conducting a similar survey for students after the end of the pandemic, the effect of companion animals and attachment to their companion animals on the mental states of students may be investigated, along with whether the current results were obtained only in the unique situation of the COVID-19 in which people’s interactions were limited, or the same results will be obtained even after the pandemic has ended.

## 5. Conclusions

This study aimed to investigate whether living with a companion animal and attachment level influence the mood states of university students majoring in animal sciences during the COVID-19 pandemic in Japan. The results indicated that companion animal ownership and having strong attachment to their companion animals would have positive effects on the mental states of students majoring in animal sciences, especially in TMD, confusion, fatigue, and vigor. However, the study population was limited in its size and by the students’ major, so the results need to be interpreted as a possible effect of companion animals, and not as conclusive evidence to support the effects of animals. The results cannot be generalized to other students who are majoring in other subjects.

## Figures and Tables

**Table 1 animals-11-02887-t001:** Demographic Characteristics of the Respondents by Gender and Companion Animal Ownership/Attachment.

		Total	Gender		Companion Animal Ownership/ Attachment		
		*n* (%)	*n* (%)	*n* (%)	*n* (%)	*n* (%)	*n* (%)
Age (years) †			Men	Women	Non-Owner	Owner with Low Attachment	Owner with High Attachment
	18	49 (27.2)	13 (23.2)	36 (29.0)	27 (30.7)	11 (25.6)	11 (22.4)
	19	56 (31.1)	14 (25.0)	42 (33.9)	27 (30.7)	14 (32.6)	15 (30.6)
	20	39 (21.7)	12 (2.4)	27 (21.8)	20 (22.7)	6 (14.0)	13 (26.5)
	21	22 (12.2)	10 (17.9)	12 (9.7)	9 (10.2)	6 (14.0)	7 (14.3)
	22	12 (6.7)	6 (10.7)	6 (4.8)	4 (4.6)	5 (11.6)	3 (6.1)
	23	2 (1.1)	1 (1.8)	1 (0.8)	1 (1.1)	1 (2.3)	0 (0.0)
Living with someone							
	Yes	128 (71.1)	40 (71.4)	88 (71.0)	48 (54.5)	37 (86.0)	43 (87.8)
	No	52 (28.9)	16 (28.6)	36 (29.0)	40 (45.5)	6 (14.0)	6 (8.6)
Perceived difficulties from the COVID-19							
	Great difficulty	22 (12.2)	9 (16.1)	13 (10.5)	12 (13.6)	4 (9.3)	6 (12.2)
	Moderate difficulty	47 (26.1)	15 (26.8)	32 (25.8)	21 (23.9)	12 (27.9)	14 (28.6)
	Slight difficulty	79 (43.9)	15 (26.8)	64 (51.6)	44 (50.0)	14 (32.6)	21 (42.9)
	No difficulty	32 (17.8)	17 (30.4)	15 (12.1)	11 (12.5)	13 (30.2)	8 (16.3)
Ways of relieving stress							
	Watching online videos	121 (67.2)	34 (60.7)	87 (70.2)	60 (68.2)	29 (67.4)	32 (65.3)
	Playing games	114 (63.3)	33 (58.9)	81 (65.3)	58 (65.9)	25 (58.1)	31 (63.3)
	Sleeping	110 (61.1)	26 (46.4)	84 (67.7)	56 (63.6)	25 (58.1)	29 (59.2)
	Checking SNS	96 (53.3)	26 (46.4)	70 (56.5)	45 (51.1)	21 (48.8)	30 (61.2)
	Playing with a companion animal	90 (50.0)	27 (48.2)	63 (50.8)	8 (9.1)	34 (79.1)	48 (98.0)
	Reading books and comics	90 (50.0)	27 (48.2)	63 (50.8)	38 (43.2)	26 (60.5)	26 (53.1)
	Making a phone call to friends	84 (46.7)	23 (41.1)	61 (49.2)	44 (50.0)	17 (39.5)	23 (46.9)
	Watching TV	75 (41.7)	19 (33.9)	56 (45.2)	27 (30.7)	19 (44.2)	29 (59.2)
	Watching movies	64 (35.6)	17 (30.4)	47 (37.9)	24 (27.3)	17 (39.5)	23 (46.9)
	Talking with family members (making a phone call to family members)	56 (31.1)	14 (25.0)	42 (33.9)	29 (33.0)	10 (23.3)	17 (34.7)
	Exercising	56 (31.1)	17 (30.4)	39 (31.5)	22 (25.0)	13 (30.2)	21 (42.9)
	Others	17 (9.4)	5 (8.9)	12 (9.7)	6 (6.8)	6 (14.0)	5 (10.2)
Companion animal ownership							
	Yes	92 (51.1)	27 (48.2)	65 (52.4)	-	43 (46.7)	49 (53.3)
	No	88 (48.9)	29 (51.8)	59 (47.6)	-	-	-
	Type of animals						
	Dog	61 (66.3)	19 (70.4)	42 (64.6)	-	20 (32.8 *)	41 (67.2 *)
	Cat	23 (25.0)	10 (37.0)	13 (20.0)	-	13 (56.5 *)	10 (43.5 *)
	Small animal	17 (18.5)	6 (22.2)	11 (16.9)	-	12 (70.6 *)	5 (29.4 *)
	Others	24 (26.1)	5 (18.5)	19 (29.2)	-	13 (54.2 *)	11 (45.8 *)
	Attachment to their companion animal						
	<128.8	43 (46.7)	17 (63.0)	26 (40.0)	-	-	-
	>128.8	49 (53.3)	10 (37.0)	39 (60.0)	-	-	-

**†** The students’ year in school was not asked in the questionnaire, but 90% of students at this university enroll immediately after graduating from high school, and accelerated promotion is not common in Japan. The estimated distribution of the students’ years is freshman (*n* = 77, 42.8%), sophomore (*n* = 47, 26.1%), junior (*n* = 37, 17.2%), senior (*n* = 25, 13.9%). * The total number of each type of animals was used as the denominator to calculate the percentage.

**Table 2 animals-11-02887-t002:** Summary of Multiple Regression Analyses for TMD and Seven Subscales of POMS2.

Dependent Variable	Independent Variable	B	SEB	β	VIF
TMD					
	Perceived difficulties from COVID-19 pandemic	3.256	0.824	0.281 ***	1
	Pet ownership/Attachment to pets	−2.052	0.881	−0.165 *	1
	Adjusted R2	0.098 ***			
(AH)					
	Perceived difficulties from COVID-19 pandemic	2.389	0.634	0.272 ***	1
	Adjusted R2	0.069 ***			
(CB)					
	Perceived difficulties from COVID-19 pandemic	2.123	0.795	0.191 **	1.001
	Pet ownership/Attachment to pets	−2.832	0.889	−0.238 **	1.096
	Number of ways to relieve stress	0.807	0.319	0.189 *	1.096
	Adjusted R2	0.088 ***			
(DD)					
	Perceived difficulties from COVID-19 pandemic	3.060	0.840	0.264 ***	1
	Adjusted R2	0.064 ***			
(FI)					
	Perceived difficulties from COVID-19 pandemic	2.431	0.897	0.195 **	1.001
	Pet ownership/Attachment to pets	−2.784	1.003	−0.209 **	1.096
	Number of ways to relieve stress	0.713	0.360	0.149 *	1.096
	Adjusted R2	0.071 ***			
(TA)					
	Perceived difficulties from COVID-19 pandemic	2.750	0.759	0.255 ***	1.005
	Gender (women)	−4.247	1.507	−0.201 **	1.034
	Number of ways to relieve stress	0.789	0.300	0.190 **	1.064
	Dog ownership	−2.975	1.474	0.144 *	1.035
	Adjusted R2	0.120 ***			
(VA)					
	Perceived difficulties from COVID-19 pandemic	−1.819	0.729	−0.181*	1.000
	Pet ownership/Attachment to pets	1.886	0.780	0.176 *	1.000
	Adjusted R2	0.055 **			
(F)					
	Gender (women)	−4.458	1.669	−0.196**	1.000
	Adjusted R2	0.033 **			

TMD: total mood disturbance. AH: anger–hostility, CB: confusion–bewilderment, DD: depression–dejection, FI: fatigue–inertia, TA: tension–anxiety, VA: vigor–activity, F: friendliness. *, **, ***: *p* < 0.05, *p* < 0.01, *p* < 0.001.

## Data Availability

The data presented in this study are available on request from the author.

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
