# Peer review of "Companion Animal Ownership and Mood States of University Students Majoring in Animal Sciences during the COVID-19 Pandemic in Japan"

_animals, 2021, doi:10.3390/ani11102887_

Round 1

Reviewer 1 Report

After careful reading I have a lot of doubts about this work. Why only animal science students were surveyed. Furthermore, the sample of 180 students is too small for a valid conclusion. What was the response rate? More information about the survey itself is necessary, what year the students are, where they come from (city, village). In the discussion section, it is quite briefly described and compared with previous similar research. The chapter conclusions claims something that has not been researched at all ... The research itself has potential, but it needs to be further adjusted and thoroughly processed. 

Author Response

We appreciate the meaningful reviewer’s comments. We edited the manuscript based on the comments.

Why only animal science students were surveyed.

These students were recruited for two reasons. First, this population made it possible to recruit enough pet owners. The overall rate of Japanese households’ pet ownership is only 28.2% (dogs: 11.9%, cats: 9.6%)and we hypothesized that university students, especially those living alone, were less likely to live with a companion animal. Therefore, we focused on the students majoring in animal sciences, as they seemed more likely to have a companion animal. Second, we recruited only students of similar school situations. The situations of other universities were not well understood at the time when this study was being prepared, because it was just after the new school year had started and when the first declaration of a state of emergency was issued. Japanese universities were in the midst of a chaotic situation. Therefore, we only focused on students from one university, with which the authors are affiliated.

However, it is a limitation that we gathered answers only from the students majoring in animal sciences. The students who have interests in animals may have positive relationship with animals and appreciate the effect of animals more often than the students with other interests. It is possible that the study results were unique to the students having interests in animals and that the results cannot be generalized to other university populations.

To make this limitation clear, we changed the title and added some sentences as below.

Title:

Companion animal ownership and mood states of university students majoring in animal sciences under the COVID-19 pandemic in Japan.

Abstract:

L29-34: This finding indicated that companion animal ownership with high attachment to their companion animals would relate to a positive mood in university students majoring in animal sciences during the COVID-19 pandemic. However, because of the limited population in its size and by the students’ major, the results need to be interpreted as a possible effect of companion animals, and not as conclusive evidence to support the effects of animals.

Material and Methods:

L104-118: The study was announced on an online bulletin board of some classes; 314 students, mainly in the first three years of their studies, had access to the bulletin board. The students come from all parts of Japan, and about half of them were living with their families. The university campus is located roughly 1–1.5 hours by train from central Tokyo. These students were recruited for two reasons. First, this population made it possible to recruit enough pet owners. The overall rate of Japanese households’ pet ownership is only 28.2% (dogs: 11.9%, cats: 9.6%) [3] and we hypothesized that university students, especially those living alone, were less likely to live with a companion animal. Therefore, we focused on the students majoring in animal sciences, as they seemed more likely to have a companion animal. Second, we recruited only students of similar school situations. The situations of other universities were not well understood at the time when this study was being prepared, because it was just after the new school year had started and when the first declaration of a state of emergency was issued. Japanese universities were in the midst of a chaotic situation. Therefore, we only focused on students from one university, with which the authors are affiliated.

Limitations:

L409-413: Finally, this survey focused only on students majoring in animal sciences, who may have positive relationship with animals and appreciate the effect of animals more often than the students with other interests. It is possible that the study results were unique to students who have an interest in animals and that the results cannot be generalized to other university populations.

Conclusions:

L422-428: The results indicated that companion animal ownership and having strong attachment to their companion animals would have positive effects on the mental states of students majoring in animal sciences, especially on TMD, confusion, fatigue, and vigor. However, the study population was limited in its size and by the students’ major, so the results need to be interpreted as a possible effect of companion animals, and not as conclusive evidence to support the effects of animals. The results cannot be generalized to other students who are majoring in other subjects.

The sample of 180 students is too small for a valid conclusion.

>This is true. We changed the sentence in the limitation and conclusion as below.

Abstract:

L29-34: This finding indicated that companion animal ownership with high attachment to their companion animals would relate to a positive mood in university students majoring in animal sciences during the COVID-19 pandemic. However, because of the limited population in its size and by the students’ major, the results need to be interpreted as a possible effect of companion animals, and not as conclusive evidence to support the effects of animals.

Limitations:

L399-400: This study was conducted voluntarily, and a small sample size was also a limitation. A larger sample size is required for a valid conclusion.

Conclusions:

L422-428: The results indicated that companion animal ownership and having strong attachment to their companion animals would have positive effects on the mental states of students majoring in animal sciences, especially on TMD, confusion, fatigue, and vigor. However, the study population was limited in its size and by the students’ major, so the results need to be interpreted as a possible effect of companion animals, and not as conclusive evidence to support the effects of animals. The results cannot be generalized to other students who are majoring in other subjects.

What year the students are.

>We did not ask the students’ year in school. However, 90% students at this university enroll immediately after graduating from high school, and accelerated promotion is not common in Japan. The estimated distribution of the students’ years was added in the newly created table.

Under the Table 1:

 The students’ year in school was not asked in the questionnaire, but 90% of students at this university enroll immediately after graduating from high school, and accelerated promotion is not common in Japan. The estimated distribution of the students’ years is: freshman (n = 77, 42.8%), sophomore (n = 47, 26.1%), junior (n = 37, 17.2%), senior (n = 25, 13.9%).

Where they come from.

> The students come from all parts of Japan. But as the results showed that about 40% of them lived with their family. It means they live close to the university (within 1.5 to 2 hours commute range to school). The university campus is located roughly 1-1.5 hours by train from central Tokyo. We added the following sentence in the materials and methods.

Materials and Methods:

L106-108: The students come from all parts of Japan, and about half of them were living with their families. The university campus is located roughly 1–1.5 hours by train from central Tokyo.

The chapter conclusions claims something that has not been researched at all.

>The sentence was removed and the sentence in the Limitations was changed.

Limitations:

L388-391:  First, the study was not designed to investigate the causality between companion animal ownership/attachment and mental states of students. The mental states of companion animal owners might be originally better than those of non-companion animal owners.

Removed from the conclusions.

” However, the causality of companion animal ownership/attachment and good mental states was unclear.”

Author Response

We appreciate the meaningful reviewer’s comments. We edited the manuscript based on the comments.

I would recommend shortening the first paragraph (L32-53) and reducing it to the most important information about the current situation of students in Japan (Period of closure of the university, type of teaching etc.).

The authors should consider placing the second paragraph of the introduction at the beginning, then describe the effect animals have on the well-being of humans and only then deal with the concrete situation of the students in Japan.

> Thank you for the helpful comments. We changed the order of the paragraphs and wrote the concrete situation of the students in Japan.

Introduction: (situation of the students in Japan)

L78-95: The COVID-19 pandemic has wreaked havoc worldwide and greatly changed people’s lives, including those of university students. A government study conducted in mid-May 2020, during the first declaration of a state of emergency (April 7–May 25, 2020), showed that 90.0% of all Japanese universities and colleges were providing only online teaching, 3.1% were providing only in-person teaching, and 6.8% were providing a hybrid of in-person and online classes [16]. No detailed research has been completed on this to date, but it was reported that many universities initiated a campus closure during the first state of emergency, which was right after the school year began (the first semester typically starts on April 1) [17]. Even in the second semester, 90.3% of the universities in the Kanto district (consisting of Tokyo and five other prefectures), where 33% of the population live, held hybrid in-person and online courses, and only 8.8% were providing in-person teaching for all classes [18]. Among the universities using a hybrid approach, 32.7% were providing classes mainly online. Although government data on the types of teaching offered by universities for the year 2021 have not yet been reported, more universities seem to have increased their ratios of in-person teaching [19]. During the COVID-19 pandemic and the subsequent great changes in teaching style, some studies have reported that university students were experiencing new burdens and psychological problems [20, 21].

L70-71: The sentence “People living with a companion animal have good mental health” should be expressed more carefully. There are already studies that question the positive effects of animals on the well-being of humans as the authors themselves state.

> We changed the sentence as below.

Introduction:

Although studies have not shown only the positive effects of living with animals, as explained later, many have indicated that people who live with a companion animal are likely to have good mental health [5, 6].

L77-79: should be removed (no connection with the present study)

>These sentences were removed.

Moreover, information about Material and Methods should be revised. For example, there are no information about how the surveyed difficulties from the COVID-19 pandemic and the ways of relieving stress were defined. Do the mentioned questions include standardized stressors and stress management skills?

> As for the perceived difficulties brought on by the COVID-19 pandemic, the students were asked one question: whether or not they were experiencing difficulties due to the effects of COVID-19, and they were asked to rate with a four-point Likert scale. The question for the ways of relieving stress is also not a standardized stress management skill. These activities were selected based on research in which students noted their preferred ways of relieving stress [23, 24] and some additional activities, such as checking SNS and playing with a companion animal. These were not standardized questions aimed at measuring the students’ stress management skills, but rather questions aimed at determining what kinds of activities the students use to relieve their stress.

We added the details in the Materials and Methods and the questionnaire as a supplemental material.

Materials and Methods:

L136-140: As for the perceived difficulties brought on by the COVID-19 pandemic, the students were asked one question: whether or not they were experiencing difficulties due to the effects of COVID-19, and they were asked to rate with a four-point Likert scale from 1, 2, 3, and 4 as no difficulty, slight difficulty, moderate difficulty, and great difficulty, respectively.

L144-149: These activities were selected based on research in which students noted their preferred ways of relieving stress [23, 24] and some additional activities, such as checking SNS and playing with a companion animal. These were not standardized questions aimed at measuring the students’ stress management skills, but rather questions aimed at determining what kinds of activities the students use to relieve their stress.

In the description of the statistical analysis, the information on the multiple comparison test is missing.

L178-184: Statistical reporting is missing.

> The missing test and an additional statistical analysis wad also added.

L169-173: A multiple comparison test with Bonferroni correction was used to determine the differences between companion ownership/attachment to their companion animals and the students’ perceived difficulties from the COVID-19 pandemic. The Mann-Whitney U test was used to check the differences in levels of attachment to a companion animal, based on gender and type of animal.

Table 1: There are typing errors in the table “Pet owenership” should be “Pet ownership”

>It was corrected.

Overall, the results are presented too unclearly.

>Thank you for the comment. We created a new table for the better understanding of the results.

Within the discussion, it would be helpful to fully name the mentioned variables (L252-L253) for a better understanding.

>Thank you for the suggestion. We wrote the full name of the variables.

L292: Check formatting

>Changed the format.

Reviewer 3 Report

I have read the manuscript very carefully. The aim of the study is interesting investigating association of pet animal ownership/attachment to pets with mood states of university students under the COVID-19 pandemic. However, I have doubts about the representativeness of the sample. I find the sample size of 180 students too small to represent the perception of the university students in the state and to draw credible study conclusions. What was the response rate? In addition, the perception of students majoring in animal science was only investigated which could have affected study results. Furthermore, the analysis on attachment to pets was performed on approximately 90 students under the study that had a pet animal. I would like to see the questionnaire used in the study, for example, in Supplementary Materials. The study year is not stated either, only that the average student age was 19.4 years old. The age of the students and the year of the study do not have to refer to the same. The results related to student demographics would be better presented in the table. The Discussion is quite poor. For example, the authors found that the gender was related to certain mood states; yet, not discussed it, i.e., compared it with similar studies. The study has too many limitations. At the end of the paper, the authors stated that the causality of companion animal ownership/attachment and good mental states was unclear. This was also not investigated.

Author Response

We appreciate the meaningful reviewer’s comments. We edited the manuscript based on the comments.

I find the sample size of 180 students too small to represent the perception of the university students in the state and to draw credible study conclusions.

>This is true. We changed the sentence in the limitation and conclusion as below.

Title:

Companion animal ownership and mood states of university students majoring in animal sciences under the COVID-19 pandemic in Japan.

Abstract:

L29-34: This finding indicated that companion animal ownership with high attachment to their companion animals would relate to a positive mood in university students majoring in animal sciences during the COVID-19 pandemic. However, because of the limited population in its size and by the students’ major, the results need to be interpreted as a possible effect of companion animals, and not as conclusive evidence to support the effects of animals.

Limitations:

L399-400: This study was conducted voluntarily, and a small sample size was also a limitation. A larger sample size is required for a valid conclusion.

Conclusions:

L422-428: The results indicated that companion animal ownership and having strong attachment to their companion animals would have positive effects on the mental states of students majoring in animal sciences, especially on TMD, confusion, fatigue, and vigor. However, the study population was limited in its size and by the students’ major, so the results need to be interpreted as a possible effect of companion animals, and not as conclusive evidence to support the effects of animals. The results cannot be generalized to other students who are majoring in other subjects.

What was the response rate?

>The study was announced on an online bulletin board for the students at the university. It was 314 students who could check the bulletin board. It was unclear whether all students who could access the online bulletin board actually read the message inviting them to participate in the survey, but if all students did, the response rate would be 57.3%.  

We added to make this clear in the Materials and Method and Results sections.

Materials and Methods:

L104-106: The study was announced on an online bulletin board of some classes; 314 students, mainly in the first three years of their studies, had access to the bulletin board.

Results:

L178-181: It was unclear whether all students who could access the online bulletin board actually read the message inviting them to participate in the survey, but if all students did, the response rate would be 57.3%.

The perception of students majoring in animal science was only investigated which could have affected study results.

>Thank you for the important comment. The results cannot be generalized to other university students and we changed the title and added sentence in the limitation as below.

Title:

Companion animal ownership and mood states of university students majoring in animal sciences under the COVID-19 pandemic in Japan.

Abstract:

L29-34: This finding indicated that companion animal ownership with high attachment to their companion animals would relate to a positive mood in university students majoring in animal sciences during the COVID-19 pandemic. However, because of the limited population in its size and by the students’ major, the results need to be interpreted as a possible effect of companion animals, and not as conclusive evidence to support the effects of animals.

Limitations:

L409-413: Finally, this survey focused only on students majoring in animal sciences, who may have positive relationship with animals and appreciate the effect of animals more often than the students with other interests. It is possible that the study results were unique to students who have an interest in animals and that the results cannot be generalized to other university populations.

Conclusions:

L422-428: The results indicated that companion animal ownership and having strong attachment to their companion animals would have positive effects on the mental states of students majoring in animal sciences, especially on TMD, confusion, fatigue, and vigor. However, the study population was limited in its size and by the students’ major, so the results need to be interpreted as a possible effect of companion animals, and not as conclusive evidence to support the effects of animals. The results cannot be generalized to other students who are majoring in other subjects.

I would like to see the questionnaire used in the study.

>The questionnaire was added in the supplement material.

The study year is not stated either, only that the average student age was 19.4 years old. The age of the students and the year of the study do not have to refer to the same.

>We did not ask the students’ year in school. However, 90% students at this university enroll immediately after graduating from high school, and accelerated promotion is not common in Japan. The estimated distribution of the students’ years was added in the newly created table.

Under the Table 1:

The students’ year in school was not asked in the questionnaire, but 90% of students at this university enroll immediately after graduating from high school, and accelerated promotion is not common in Japan. The estimated distribution of the students’ years is: freshman (n = 77, 42.8%), sophomore (n = 47, 26.1%), junior (n = 37, 17.2%), senior (n = 25, 13.9%).

The results related to student demographics would be better presented in the table.

>Thank you for the suggestion. We created a new table for the better understanding of the student demographics.

The discussion is too poor. For example, the authors found that the gender was related to certain mood states; yet, not discussed it, i.e., compared it with similar studies.

>Thank you for the useful comment. We checked the results again and we found gender differences on their attachment to their companion animals. Based on the result, we added some discussion. We also added some discussion for the number of ways to relieve stress.

Results:

L199-207: The Mann-Whitney U test showed that the attachment score was significantly higher among women than men (women: mean = 132.5, S.D. = 22.1; men: mean = 119.9, S.D. = 22.1, p < 0.05, r = 0.25). When the type of animal was indicated in detail, a gender difference was only revealed among dog owners (women: mean = 141.4, S.D. = 17.9; men: mean = 126.2, S.D. = 16.6, p < 0.01, r = 0.38). Similarly, dog owners scored significantly higher for attachment, compared to other pet owners among women (dog owners: mean = 141.4, S.D. = 17.9; other pet owners: mean = 116.3, S.D. = 20.0, p < 0.01, r = 0.54). No statistical difference was shown among men (dog owners: mean = 126.2, S.D. = 16.6; other pet owners: mean = 105.0, S.D. = 27.3, p = 0.06, r = 0.36).

Discussions:

L332-385: The results showed some gender differences. Women scored significantly higher in terms of attachment to their companion animals, especially dogs, when compared to men. This result is consistent with Winefield and her colleagues’ research, which also showed higher pet attachment among women than men, and the score was significantly higher for dog owners than for cat or other pet owners [30 ]. Although previous research on gender and pet attachment has been inconsistent, those studies reporting a relationship between gender and pet attachment always show that women have higher levels of attachment to pets than men do [31-33]. As discussed later, interactions with companion animals, espe-cially dogs, might have served an important role for women under the limited social in-teractions allowed by the pandemic-related lockdown.

Gender also served as a predictor variable on F (friendliness) and TA (ten-sion-anxiety). Gender was the only predictor variable, and women were more likely to have worse scores in F. The questions for F consisted of interpersonal items. Because peo-ple were required to engage in social distancing because of the COVID-19 pandemic, in-person social interaction decreased greatly. Previous studies have reported that women have larger social networks and receive support from more sources than men [34], and they were more likely to have friends or family to whom they could talk about their health. Women were also more likely to be a member of one or more community organizations [35]. Women might have been more affected by the great changes in social interaction than men were because of the COVID-19 pandemic, as shown in a study conducted in the UK wherein women reported greater degrees of loneliness than men during the pandemic [36].

As for TA, the women’s better scores in TA, compared to men, were the opposite of other findings on mental health during the COVID-19 pandemic. It was also a surprising result to consider the worse F scores reported in women. Studies conducted in several countries, including Japan, have shown that being female is one of the risk factors for poor mental health [37, 38]. One study focusing on medical students in Japan did not show gender differences in mental health [39]; however, more studies have indicated being fe-male is a mental health risk factor. One possible reason for the current result might be ex-plained by women’s higher levels of attachment to their dogs, as “dog ownership” was another predictor variable for TA. Having a positive relationship with a dog might have worked to maintain women’s mental health, even during the COVID-19 pandemic. Many studies have investigated the relationship between pet ownership and mental health, but there is only limited research specifically investigating the relationship between dog own-ership and anxiety. One study showed that dog ownership relates to less anxiety [40], while another showed no relationship between dog ownership and anxiety [41]. These works did not investigate in detail the relationships people have with dogs. For example, one study on people diagnosed with fibromyalgia, by Silva and colleagues, reported that dog ownership was associated with reduced levels of anxiety in individuals who also have moderate to high levels of human social support, but the opposite was true for indi-viduals with low levels of social support [42]. This study also showed that some interac-tions with dogs, such as petting and stroking the dog, which help to manage pain, are as-sociated with a significant reduction in anxiety levels. This was so regardless of levels of human social support. Similarly, a study on older adults showed that dog ownership was not associated with anxiety levels, but the frequency of a dog’s presence was moderately negatively related to anxiety [43]. Based on these results, dog owners who have a positive relationship with their dogs may have less anxiety.

Finally, the number of ways to relieve stress served as predictor variable in FI (fa-tigue–inertia), CB (confusion–bewilderment), and TA, and these scores were worse when students used more activities to relieve stress. Even during the pandemic, students had to participate in classes and hand in more reports than before the pandemic, which caused a great deal of stress [44]. Self-management was required of the students to adjust to the changes. In this situation, students who chose to engage in many activities to relieve stress may not have had an orderly life, or may not have been able to focus on their schoolwork, which would have led to negative mental moods.

At the end of the paper, the authors stated that the causality of companion animal ownership/attachment and good mental states was unclear. This was also not investigated.

>The sentence was removed and the sentence in the Limitations was changed.

Limitations:

L388-391: First, the study was not designed to investigate the causality between companion animal ownership/attachment and mental states of students. The mental states of companion animal owners might be originally better than those of non-companion animal owners.

Round 2

Reviewer 1 Report

Althought paper is now rewritten I find again some mistakes that must be, if Editor decide to accept paper, improved (changed).

Sentences from 57 to 66 line can be rewritten because it all consider social interactions....

Line 123 is repetitions of sentence in line 103. So sentence in line 103 can be:

University students majoring in animal sciences at.....were surveyed.

Line 139- Likert scale 1,2,3,4....is completely differently described than in the Questionary.

What does it mean SNS (line 142). Is it maybe SMS?

Again response rate is bady described. Line 180... response rate in this survey was 57,3%, not would be.

Line 248 - AH score did not include companion animal ownership/attachment is it can be seen in Table 2.

Line 293- Except F not except FI

Line - 339 - explain why women have higher leveles of attachment to pets....

References must be written according to journal style....

Examples of the good related investigations and well written papers can be:

Mikuš, T.; Ostović, M.; Sabolek, I.; Matković, K.; Pavičić, Ž.; Mikuš, O.; Mesić, Ž. Opinions towards Companion Animals and Their Welfare: A Survey of Croatian Veterinary Students. Animals 2020, 10, 199. https://doi.org/10.3390/ani10020199

Ostović, Mario & Mikuš, Tomislav & Pavicic, Z & Matković, Kristina & Mesic, Zeljka. (2017). Influence of socio-demographic and experiential factors on the attitudes of Croatian veterinary students towards farm animal welfare. Veterinární medicína. 62. 2017-417. 10.17221/172/2016-VETMED.

Author Response

We appreciate the meaningful reviewer’s comments. We edited the manuscript based on the comments.

Sentences from 57 to 66 line can be rewritten because it all consider social interactions....

Thank you for the comment. We added sentence to include reports other than social interaction.

The inconsistency of benefits from animals is also reported among other known benefits, such as those related to physical and mental health and pet ownership, including cardiovascular disease, depression and anxiety [14, 15]

Line 123 is repetitions of sentence in line 103. So sentence in line 103 can be:

University students majoring in animal sciences at.....were surveyed.

We changed the sentence as suggested.

Line 139- Likert scale 1,2,3,4....is completely differently described than in the Questionary.

The translated questionnaire was wrong. We uploaded the revised questionnaire.

What does it mean SNS (line 142). Is it maybe SMS?

It is a social networking service. We added the full name.

Again response rate is bady described. Line 180... response rate in this survey was 57,3%, not would be.

The sentence was simplified as “the response rate in this survey was 57.3%”

Line 248 - AH score did not include companion animal ownership/attachment is it can be seen in Table 2.

It was mistake. “companion animal ownership/attachment” was deleted from the sentence.

Line 293- Except F not except FI

We corrected FI to F.

Line - 339 - explain why women have higher leveles of attachment to pets....

We added the sentences below.

Interpreting the results of a survey conducted in Japan, Sugita reported that women spent more time with their pets than men. Furthermore, a positive correlation between attachment to dogs and the time spent with dogs was shown for both men and women. In the present study, the time spent with pets was not asked; however, it is possible that female students spent more time with their pets, and as a result, they might have been more attached to their pets in comparison to male students [36, 37]

References must be written according to journal style....

The references were rewritten.

Reviewer 3 Report

The manuscript has been improved. However, I still have some remarks.

line 12 – I would say regarding their demographic data (age and gender were only some of the variables investigated)

lines 104 and 122 – there is no need to say that survey was voluntary at two places

line 139 – regarding evaluation of perceived difficulties the authors have written that they used 4-point Likert scale, whereby 1 means no difficulty, etc. However, in the Supplementary Materials (Q5-1), 1 stands for great difficulty, etc. Is this a mistake or ...?

line 169 – the authors should state the statistical software used

lines 178-181 – this sentence is not clear. What does this mean ... the response rate would be?

line 249 – companion animal ownership/attachment was not correlated with AH, as presented in Table 2

line 293 – except F, not FI, as presented in Table 2

line 440 – please write all references according to the journal style

Author Response

We appreciate the meaningful reviewer’s comments. We edited the manuscript based on the comments.

line 12 – I would say regarding their demographic data (age and gender were only some of the variables investigated)

It was changed as suggested.

lines 104 and 122 – there is no need to say that survey was voluntary at two places

We changed the first sentence as below.

“University students majoring in animal sciences at Teikyo University Science were surveyed.”

line 139 – regarding evaluation of perceived difficulties the authors have written that they used 4-point Likert scale, whereby 1 means no difficulty, etc. However, in the Supplementary Materials (Q5-1), 1 stands for great difficulty, etc. Is this a mistake or ...?

The translated questionnaire was wrong. We uploaded the revised questionnaire.

line 169 – the authors should state the statistical software used

The software was named in the statistics section.

lines 178-181 – this sentence is not clear. What does this mean ... the response rate would be?

The sentence was simplified as “the response rate in this survey was 57.3%”

line 249 – companion animal ownership/attachment was not correlated with AH, as presented in Table 2

It was mistake. “companion animal ownership/attachment” was deleted from the sentence.

line 293 – except F, not FI, as presented in Table 2

We corrected FI to F.

line 440 – please write all references according to the journal style

The references were rewritten.